# Will Absolute Risk Estimation for Time to Next Screen Work for an Asian Mammography Screening Population?

**DOI:** 10.3390/cancers15092559

**Published:** 2023-04-29

**Authors:** Peh Joo Ho, Elaine Hsuen Lim, Nur Khaliesah Binte Mohamed Ri, Mikael Hartman, Fuh Yong Wong, Jingmei Li

**Affiliations:** 1Genome Institute of Singapore (GIS), Agency for Science, Technology and Research (A*STAR), 60 Biopolis Street, Genome, Singapore 138672, Singapore; hopj@gis.a-star.edu.sg; 2Saw Swee Hock School of Public Health, National University of Singapore and National University Health System, Singapore 117549, Singapore; 3Department of Surgery, Yong Loo Lin School of Medicine, National University of Singapore and National University Health System, Singapore 119228, Singapore; 4Division of Medical Oncology, National Cancer Centre Singapore, Singapore 168583, Singapore; 5Department of Surgery, University Surgical Cluster, National University Hospital, Singapore 119228, Singapore; 6Division of Radiation Oncology, National Cancer Centre Singapore, Singapore 168583, Singapore

**Keywords:** breast cancer, Gail model, calibration, discriminatory ability, absolute risk

## Abstract

**Simple Summary:**

Personalized breast cancer screening has the potential to improve the accuracy and effectiveness of breast cancer screening by tailoring the screening protocol to an individual’s risk factors. Currently, most women undergo mammograms at regular intervals based on age and family history, regardless of their risk factors. We examined the applicability of the Gail model, a well-known breast cancer prediction tool including several risk factors, in an Asian population. While the tool predicts the risks of developing breast cancer in the next 2, 5, 10, and 15 years to some extent, the age at which the disease occurs cannot be estimated. In addition, the tool performs better for longer prediction horizons (10 or 15 years), limiting its utility for risk prediction for two-year screening intervals.

**Abstract:**

Personalized breast cancer risk profiling has the potential to promote shared decision-making and improve compliance with routine screening. We assessed the Gail model’s performance in predicting the short-term (2- and 5-year) and the long-term (10- and 15-year) absolute risks in 28,234 asymptomatic Asian women. Absolute risks were calculated using different relative risk estimates and Breast cancer incidence and mortality rates (White, Asian-American, or the Singapore Asian population). Using linear models, we tested the association of absolute risk and age at breast cancer occurrence. Model discrimination was moderate (AUC range: 0.580–0.628). Calibration was better for longer-term prediction horizons (E/O_long-term ranges_: 0.86–1.71; E/O_short-term ranges_:1.24–3.36). Subgroup analyses show that the model underestimates risk in women with breast cancer family history, positive recall status, and prior breast biopsy, and overestimates risk in underweight women. The Gail model absolute risk does not predict the age of breast cancer occurrence. Breast cancer risk prediction tools performed better with population-specific parameters. Two-year absolute risk estimation is attractive for breast cancer screening programs, but the models tested are not suitable for identifying Asian women at increased risk within this short interval.

## 1. Introduction

Breast cancer five-year survival of over 80% is expected with detection at the early stages [1,2,3]. An effective strategy for mortality reduction is to routinely screen asymptomatic women using mammography from the age of 50 years [4]. The emphasis is on “routine”, as noncompliance with scheduled screens lowers the benefits that mammography screening can provide to both the individual and the healthcare system [5]. The involvement of individuals in the decision to adhere to screening with the help of personalized risk-benefit assessments has shown promising results in increasing screening compliance rates [2,5,6].

In terms of cost-effectiveness, age-targeted regular mammography screening for breast cancer is reported to be cost-effective, even when considering false positives and overdiagnosis [7]. However, considering a woman’s individual breast cancer risk factors can potentially improve the cost-effectiveness of mammography screening [8]. Breast cancer risk prediction models thus present an opportunity in nationwide risk-based breast cancer screening programs by identifying individuals at high risk who may benefit more from interventions [9,10,11].

Current risk assessment tools estimate absolute risks for a time horizon of at least five years [12,13,14,15,16]. Eriksson et al. highlighted that a risk model for effective screening should evaluate the risk of breast cancer in the time between the recently completed screen and the following planned screen to identify women who require additional screening during the interval [17,18]. Mammography screening for breast cancer intervals ranges from one to three years [19,20]. When compared to employing a five-year risk assessment, a two-year absolute risk prediction might lead to fewer high-risk women receiving false positives and being sent for further screening. A five-year risk assessment, on the other hand, could be too brief for use in primary prevention given that breast cancer development can take longer than five years [17].

The efficacy of population-based screening programs for breast cancer also depends on accurate risk assessment. Prediction models associated with high sensitivity and specificity would allow programs to focus screening efforts on high-risk individuals while limiting the overtreatment of other populations. However, before a model is used to predict risk, we have to show that the model is good at predicting risk by demonstrating accuracy, reliability, and generalizability [21]. There are two main ways to tell if a mathematical model does a good job of predicting the outcome: calibration and discrimination [21,22].

The Gail model is a widely used breast cancer risk prediction model due to its simplicity. Self-assessments can be made by responding to a questionnaire [23]. The model has been reported to perform well in populations of European ancestry [24]. However, it is not well-calibrated for Asian populations [24,25]. Model calibration depends on population-specific disease and mortality trends [26]. Differing breast cancer incidence rates and mortality rates between populations have resulted in inaccurate risk predictions in Asian populations using models developed in the West [24,25,27,28].

The risk threshold at which women are identified to be at high risk of developing breast cancer determines what kind of personalized follow-up interventions are recommended for screening and prevention [29]. A five-year predicted breast cancer risk of over 1.66% was considered high enough to warrant tamoxifen chemoprevention [30]. While the absolute risk of developing breast cancer in the next five years for an average 50-year-old Caucasian woman recommended for starting mammography screening is 1.3% [30], some younger women below age 50 may already have personal breast cancer risk prediction estimates above this risk threshold, suggesting that preventive or early detection interventions, by mammography or other screening modalities, should start earlier for them [31,32]. Due to varying breast cancer incidence rates across populations, the threshold for identifying women at high risk for developing breast cancer is expected to be different.

In this study, we assessed the calibration of the Gail model in a prospective cohort of 28,234 asymptomatic Asian women undergoing mammography screening. Two-, five-, ten-, and 15-year absolute risks were calculated and evaluated. In addition, this work seeks to identify relevant thresholds for classifying Asian women as high risk for developing breast cancer.

## 2. Materials and Methods

### 2.1. Study Population

The Singapore Breast Screening Project (SBSP) was created to test the effectiveness of mammography screening on women aged 50–64 years. It was a randomized trial that invited 69,473 (41.7%) women to participate in a single free screening [33]. As part of this prospective cohort, 28,234 were recruited between October 1994 and February 1997. Passive follow-up on the cohort was performed using national registries. To study the occurrence of breast cancer after the initial screen, we excluded 34 women with a prior diagnosis of breast cancer and 62 women whose breast cancer was detected within 6 months of recruitment (i.e., breast cancer potentially detected at the first screen). Women aged above 64 years (*n* = 1744) and with unknown age at the point of screening (*n* = 14) were excluded. A total of 26,380 women were included in this study.

### 2.2. Ethics Statement

This study was approved by the SingHealth Centralised Institutional Review Board (CIRB Ref: 2019/2210) and the A*STAR Institutional Review Board (A*STAR IRB Ref:2021-141). Informed consent was not obtained as we used existing data of participants recruited from 1994 to 1997 for the Singapore Breast Screening Project (SBSP) and some participants would have died.

### 2.3. Outcome of Interest

The first diagnosis of invasive breast cancer, occurring at least 6 months after recruitment and before age 80 years, was identified via linkage with the Singapore Cancer Registry with the latest date of occurrence set at 31 December 2019.

### 2.4. Risk Factors

The participants provided information about their demographics and breast cancer risk factors through a structured questionnaire administered at recruitment. Risk factors included—the age at study entry (years), ethnicity (Chinese, Malay, Indian, other), body mass index at study entry (BMI, kg/m^2^), age at menarche (years), age at menopause (years), age at first live birth (nulliparous, <20, 20 to 24, 25 to 29, ≥30 years), parity (nulliparous, 1 to 2, ≥3), prior biopsy (yes, no), and family history of breast cancer. Women who underwent biopsy during screening were assigned a code of 1 for the risk factor of biopsy. Those who did not have a biopsy done in the past year were assigned a code of 0. Family history of breast cancer was determined by whether the woman’s mother, sister(s), or daughter(s) had a history of the disease, with a code of yes or no. The number of first-degree relatives with breast cancer was coded as 0 for no family history, 1 for having one relative with the disease (mother, sister, or daughter), and 2 for having two first-degree relatives with the disease. Recall status (yes/no) from the first screen for any reason (including abnormal mammogram, dense breast, and unclear images) was extracted from the radiology report. Missing values for BMI were replaced by the mean. Missing values for risk factors were coded as the reference level to obtain the absolute risks estimated by the Gail model [23].

### 2.5. Gail Model Absolute Risks

Absolute risks for developing breast cancer were predicted using the BCRA package in R, for the next 2, 5, 10, and 15 years. In addition, lifetime absolute risk by age 80 will be estimated (i.e., assuming all women lived to age 80 for this prediction) [23,34]. We used Gail model relative risk estimates from (1) the White population [i.e., the estimates from White 1983-87 SEER rates], and (2) the closest Asian populations [i.e., the estimates from Chinese American for our Chinese women and the estimates from other Asian for our non-Chinese women]. Corresponding incidence and mortality rates for White and Asian populations will be used to estimate the absolute risk. In addition, we obtained the absolute risk by replacing the breast cancer incidence and mortality rates of the Asian population with the rates in Singapore’s population: ethnic-specific or overall breast cancer incidence rates (period 2013 to 2017) for Singapore Citizens, and mortality rates (the year 2016) in Singapore [35].

### 2.6. Statistical Analysis

Associations between risk factors and breast cancer development before age 80 were studied using the Chi-square test for categorical variables and the Kruskal-Wallis test for continuous variables.

Calibration was studied using the ratio of the expected number of cases (E) to the observed number of cases (O). A ratio above one implies that the model overestimates the absolute risk. The 95% confidence interval will be estimated by use of the exact theory under the assumption that the observed number of cases has a Poisson distribution. The expected number of cases is divided by the lower- and upper-bound of the observed number of cases to get the 95% confidence interval. In addition, we attempted to identify subgroups of the population for which the Gail model is better calibrated. Due to an insufficient number of cases, subgroup analyses were performed for 5-, 10-, and 15-year absolute risk.

The discriminatory ability of the Gail model absolute risk prediction was assessed by the area under the receiver operating characteristic curve (AUC) calculated using the pROC library in R [36]. Breast cancer occurrence within the period specified was used as the outcome, e.g., to evaluate the AUC for the 5-year absolute risk (continuous variable), cases were the women who developed breast cancer within five years from study entry, occurrences of breast cancer at five years or later were considered women without breast cancer. We assessed model calibration and discriminatory ability in subsets of women based on breast cancer risk factors. In addition, we studied the association between absolute risk and age at breast cancer occurrence using linear models.

To determine potential thresholds for high-risk classification, absolute risk curves (2-, 5-, 10-, and 15-year) by deciles were plotted. Deciles were based on the distribution of the relative risk observed in the population at the age of entry. When we observed the same relative risk for consecutive deciles, we presented the curve for the highest decile. We report the thresholds based on the absolute risk of the 6th decile (i.e., 60th percentile) at the age of 50 years.

Analysis was done using R-version 3.6.1.

## 3. Results

Of the 26,380 women eligible for our study, 1000 (4%) developed breast cancer before the age of 80 years (Table 1). The median age at recruitment was 57 (interquartile range [IQR]: 54 to 61) years. Less than 3% (*n* = 689) of our participants reported at least one first-degree relative with breast cancer. The majority (84%) were Chinese, 6% Malays, 5% Indians, and 5% were of other ethnicities. Eighty-nine percent of our population were postmenopausal at recruitment. Ninety-three percent (*n* = 24,445) had at least one child, of which 20,245 women had three or more children. At the first mammography screening (i.e., part of our study), 1992 (8%) were recalled for follow-up but were not diagnosed with breast cancer (i.e., false positive).

### 3.1. The Gail Model Is Not Well Calibrated for Singapore’s Population

The Gail model overestimates (E/O ranges from 1.67 to 3.36) the absolute risk (2-, 5-, 10-, and 15-year projections) of developing breast cancer when we applied the relative risk estimates, incidence, and mortality rates for Whites to our population (Figure 1 and Appendix A) [23,34]. Applying the relative risk estimates, incidence and mortality rates for Asians resulted in an overestimation of the absolute risk of developing breast cancer in the next 2 to 5 years, but an underestimation of the absolute risk of developing breast cancer in the long term (i.e., 10- or 15-year absolute risk) (Figure 1 and Appendix A).

### 3.2. Improved Calibration Using Population-Specific Breast Cancer Incidence and Mortality Rates

Using breast cancer incidence and mortality rates specific to the Singaporean population, as opposed to using published rates for “Whites”, the overestimations of absolute risks were observed to be smaller (Figure 1). The highest overestimation was observed for 2-year absolute risk (E/O = 2.57, 95% CI: 1.86 to 3.55). On the contrary, the lowest overestimation was observed for lifetime absolute risk (E/O = 1.17, 95% CI: 1.08 to 1.27). Calibration of the Gail model by deciles of absolute risks presented overestimations that were generally not statistically significant for 10- and 15-year absolute risks (the 95% confidence intervals overlap the diagonal line of the calibration plot, the last row of column C in Appendix A and Figure 2, respectively). However, the overestimations remained significant when we predicted the 2- and 5-year absolute risks (Figure 3 and Appendix A, respectively).

### 3.3. Calibration of the Gail Model Differs by Risk Factor Subgroups

Most models overestimate the risk of breast cancer, regardless of the population parameters used (Appendix A). Model calibration was observed to improve when we increased the length of time considered for absolute risk (5-, 10-, and 15-year projections) (Appendix A). We observed exceptions of underestimated risk for the (1) youngest age group (40–49 years), (2) women with two or more known first-degree relatives with breast cancer, (3) women with positive recall status, and (4) women who had a prior breast biopsy. Furthermore, the Gail model overestimates the risk of breast cancer for underweight women, with E/O ranging from 1.5 to 3.9.

### 3.4. Risk Prediction Performance Was Better for Longer Time Horizons

The Gail model for the 15-year absolute risk prediction was the best-performing model, in terms of absolute risk distribution, discriminatory ability, and calibration (Figure 2). This model used the Asian-American relative risk estimates for the Gail model parameters and the breast cancer incidence rates and mortality rates of Singapore. We observed good model calibration using both the Asian-American and Singaporean incidence and mortality rates. However, the Asian-American rates systematically underestimate the risk (i.e., fall above the diagonal line in Figure 3) while Singapore’s rate systematically overestimates the risk. Short-term (2- and 5-year) absolute risk prediction was poor, even when we used the same parameters as the Gail model for 15-year estimates (Figure 3 and Appendix A). However, it should be noted that only 37 and 151 breast cancer cases were diagnosed within the first two and five years, respectively.

### 3.5. Gail Model Absolute Risk Does Not Predict the Age of Breast Cancer Occurrence

Higher absolute risk was significantly associated with an earlier occurrence of breast cancer (*p*-value < 0.05), using the Asian (beta_2-year_ = −7.82, beta_5-year_ = −2.59, beta_10-year_ = −1.36, beta_15-year_ = −0.88) or Singapore (beta_10-year_ = −0.47 and beta_15-year_ = −0.43) prediction models (Table 2). These associations were attenuated (beta value range: −0.32 to 0.15) and no longer significant when we adjusted for age at screening and ethnicity. In contrast, a higher absolute risk was associated with a later occurrence of breast cancer when we used the unadjusted White’s prediction model (beta_10-year_ = 0.52). However, the relationship was reversed and no longer significant when we adjusted for age at screening and ethnicity (beta_10-year_ = 0.14) (Table 2).

### 3.6. Rethinking What Is Considered High Risk in Asia

The absolute risk of developing breast cancer (6th decile) for a 50-year-old woman (recommended age to start subsidized mammography screening in Singapore) ranges from 0.25 to 0.39% for 2-year, 0.62–0.98% for 5-year, 1.46–2.14% for 10-year, and 2.16–3.50% for 15-year projections (Table 3). The difference between the highest and lowest threshold based on the different population parameters used (White, Asian-American, Singapore) was between 32% (10-year absolute risk) and 38% (15-year absolute risk) (Table 3). Using Singapore’s population parameters, the threshold was between that of Whites and Asian-Americans, at 0.33% for 2-year, 0.82% for 5-year, 1.74% for 10-year, and 2.79% for 15-year calculations (Table 3).

## 4. Discussion

Using a large dataset comprising 28,234 asymptomatic Asian women living in Singapore who attended a mammography screening, we showed that using population-specific breast cancer incidence and mortality rates improved the performance of the Gail model. Our results suggest that breast cancer screening should be a long-term commitment as (1) current prediction models for a shorter time horizon do not perform optimally, and (2) a higher predicted risk of developing breast cancer does not necessitate an earlier diagnosis. Breast cancer risk prediction was most well-calibrated with the highest discriminatory ability in the long-term (10- or 15-year absolute risk). The absolute risks of developing breast cancer were 0.82%, 1.74%, and 2.79% in the next five, ten, and fifteen years, respectively, for an average 50-year-old Asian woman living in Singapore (the age-specific incidence rates used for this prediction are reported in Singapore Cancer Registry 50th Anniversary Monograph, Appendix C2 [35]).

A previous work by Chay et al. on the same cohort reported good calibration in the 10-year absolute risk model (the Gail model) when they used the relative risk of the closest ethnic-specific data (Asian American Breast Cancer Study model) [37]. On the contrary, their results showed that the Gail model overestimates the risk of developing breast cancer in the next five years [37]. However, the distributions of dietary, lifestyle, and reproductive factors are different between Asian women living in Asia and Asian women living in America [38,39,40]. For example, women in Asia tend to have lower body mass index, have more children, are younger when they have their first child, start menstruation at an older age, have an earlier onset of menopause, and have lower uptake of hormone replacement therapy [38,39,40]. Hence, in our study, we examined the calibration of the Gail model in subsets of women with different risk factors and observed that the overestimation of the 5-year absolute risk was more prominent in women whose demographics were likely less represented in the Asian American Breast Cancer Study (i.e., age at menarche 14 years or older, age at menopause younger than 50 years, three or more children). We also found that the Gail model underestimates breast cancer risk for women with two or more first-degree relatives diagnosed with breast cancer and women who ever had positive recall status. Our results are in agreement with previous findings where others have shown that the Gail model cannot give the majority of women with high and moderate breast cancer risk a correct risk assessment [41,42]. Risks for women with breast cancer risk factors, such as nulliparity, late age at first birth, and early menarche, are often underestimated [41,42]. As the Gail model only considers first-degree relatives and ignores age at disease onset, the estimated risk may be underestimated risk in half of the families with cancer on the father’s side of the family [41,43,44].

Gao et al. showed in the same dataset that the performance of the Gail model in estimating 5-year absolute risk improved with the use of Singaporean race-specific invasive breast cancer rates and mortality rates for deaths not due to breast cancer [41,45]. With a longer follow-up time, we were able to extend the work to explore absolute risk predictions for longer time horizons. Similar to the work by Chay et al., we observed good calibration in long-term risk prediction (10 and 15 years) using updated Singaporean estimates of incidence and all-cause mortality rates [37]. However, changing the incidence and mortality rates did not improve model performance in the shorter term (i.e., 2- and 5-year). As the interval for mammography screening is between one to three years, with two years being the most common, breast cancer risk prediction will bring the most value if the assessments can inform women on how often they should screen [17,20,46]. Good prediction accuracy for short-term risk prediction (<3 years) has been reported for other diseases [47,48,49]. However, these models are built for individuals already at high risk. For example, the chronic kidney disease model was designed for patients with type 2 diabetes, the diabetes model for patients with prediabetes, and the cervical precancer model for individuals infected by human papillomavirus [47,48,49]. While the Gail model appears to identify women at elevated risk of developing breast cancer over a longer period, additional tools that can provide information to screeners on when breast cancer will develop after a high-risk assessment will be the Holy Grail.

Another desirable function of risk prediction models is to help inform when to start screening. Worldwide, mammography screening recommendations are based on age [20,46]. There is a significant degree of similarity in screening guidelines for women aged 50 to 69 years in various Western countries, with biennial screening every two years being the recommended approach. However, there are a few exceptions. For instance, the United Kingdom recommends a screening interval of three years, while the American College of Radiology advises an annual screening interval [46]. For women aged 40 to 49 years, screening recommendations are less consistent across different countries, and many do not have any specific guidelines. Sweden is an exception, where younger women are recommended to undergo biennial mammograms, similar to their older counterparts [46]. In the United States, selective annual mammograms are recommended for this age group [46]. A larger proportion of Asian breast cancers are, however, diagnosed at an earlier age [20,31,50,51,52]. Population-based mammography screening programs in countries such as Japan and South Korea recommend biennial screening starting from age 40 [53,54]. In Singapore, ~30% of women diagnosed with breast cancer are younger than the subsidized screening age of 50 years (National Registry of Diseases Office) [35]. Women aged 40 to 49 years are advised to consult their healthcare practitioners on the benefits and harms of getting a mammogram (Singapore Cancer Society) [55].

Instead of a one-size-fits-all age-based approach, individualized breast cancer risk predictions may aid in shared decision-making with clinicians on whether it is helpful to start screening earlier. A multi-modal approach is needed to maximize the utility of the risk-based screening strategy. For example, The BREAst screening Tailored for HEr (BREATHE) study, a prospective cohort study in Singapore, endeavors to change the current age-based screening paradigm in a prospective study for women aged 35–59 years [56]. Briefly, the initiative uses validated breast cancer risk calculators incorporating genetic (breast cancer polygenic risk score and BOADICEA breast cancer predisposition genes) and non-genetic risk factors (Gail model, mammography density, recall status) to generate individual breast cancer risk profiles [56]. Breast specialists see women at above-average risk of developing breast cancer at the participating sites to discuss breast health and interventions for early breast cancer detection. Women who are currently in the “grey zone” (<50 years) of the national screening guidelines (not targeted by current health promotion efforts) may benefit from an awareness of their personal risk profiles and be better informed to make decisions [56]. Nonetheless, the decision has to weigh the downsides that include overdiagnosis, potentially more aggressive treatment, or unnecessary anxiety due to false positives associated with mammography screening as it is uncertain if and when cancer will develop [57,58,59,60,61].

According to data from the Surveillance, Epidemiology, and End Results (SEER) registry (2017–2019, accessed on 30 August 2022, the average risk of a 50-year-old non-Hispanic White woman developing breast cancer is 2.2%, increasing to 3.4%, 4.6%, and 6.1% for the next 5, 10, and 15 years (i.e., 5-, 10-, and 15-year absolute risks of 1.2%, 2.2%, and 3.9%, respectively) [62]. The corresponding absolute risk for a 50-year-old non-Hispanic Asian or Pacific Islander is 2.2%, increasing to 3.2%, 4.4%, and 5.6% for the next 5, 10, and 15 years (i.e., 5-, 10-, and 15-year absolute risks of 1%, 2.2%, and 3.4%, respectively). In comparison, data from SBSP revealed 0.82% for 5-year, 1.74% for 10-year, and 2.79% for 15-year absolute risk projections for an average 50-year-old Singaporean woman eligible for subsidized mammography screening. According to the Gail model, a five-year absolute risk of 1.67% is considered high risk [63]. The lower values in our Asian population raise the issue of whether risk-based breast cancer screening programs in Asia should follow the guidelines of what is considered high-risk in non-Asian populations.

The strengths and limitations of using SBSP to evaluate the Gail model have been elaborated by Chay et al. and Gao et al. [41]. The source of this study is a large population-based randomized trial on mammography screening in an Asian population [64,65]. All residents in Singapore are assigned a unique National Registration Identity Card (NRIC) number, which facilitates accurate linkage to the Singapore Cancer Registry. Data from healthcare providers are used to achieve comprehensive cancer registration. Clinical medical data were used to verify all cancer notifications. Overall completeness is estimated to be above 97% [66]. However, the response rate for study enrolment was low at 41.7% and the number of breast cancer cases observed even with the longer follow-up is modest. The amount of ascertainment of incident cases is a source of concern, in this case, Ng et al. showed that the prevalence ratios in this study population were similar to other randomized breast cancer screening studies in Western populations [33]. In addition, recruitment for SBSP took place in 1994, nearly three decades ago [33]. During this time, the age-standardized incidence rate for breast cancer has increased ~1.7 times from 43.6 per 100,000 population (1993–1997) to 72.6 (2015–2019) (Singapore Cancer Registry Annual Report 2019) [35]. The population may also have adopted Western lifestyles to a larger extent [67,68].

## 5. Conclusions

Breast cancer risk prediction tools perform better with population-specific parameters. Two-year absolute risk estimation is attractive for breast cancer screening programs, but current models are not tailored to identify women at increased risk within this short interval. The importance of routine screening should be emphasized, as it is not currently possible to predict the age of disease onset. Risk-threshold-dependent recommendations for interventions will need to be reconsidered for specific populations.

## Figures and Tables

**Figure 1 cancers-15-02559-f001:**
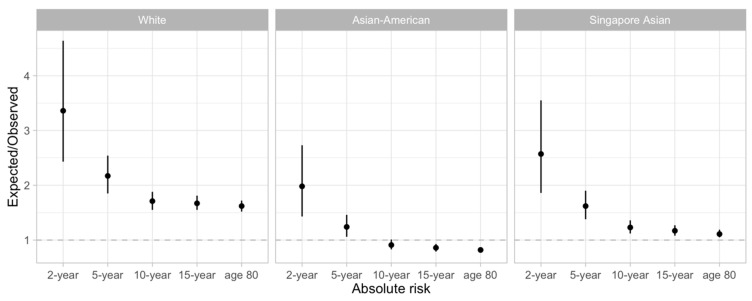
Calibration of the Gail model using the relative risk estimates from the White population [White], and the closest Asian populations [Asian-American and Singapore Asian]. The BCRA package’s breast cancer incidence rates and mortality rates were replaced with the rates from Singapore’s population for the right-most panel [Singapore Asian].

**Figure 2 cancers-15-02559-f002:**
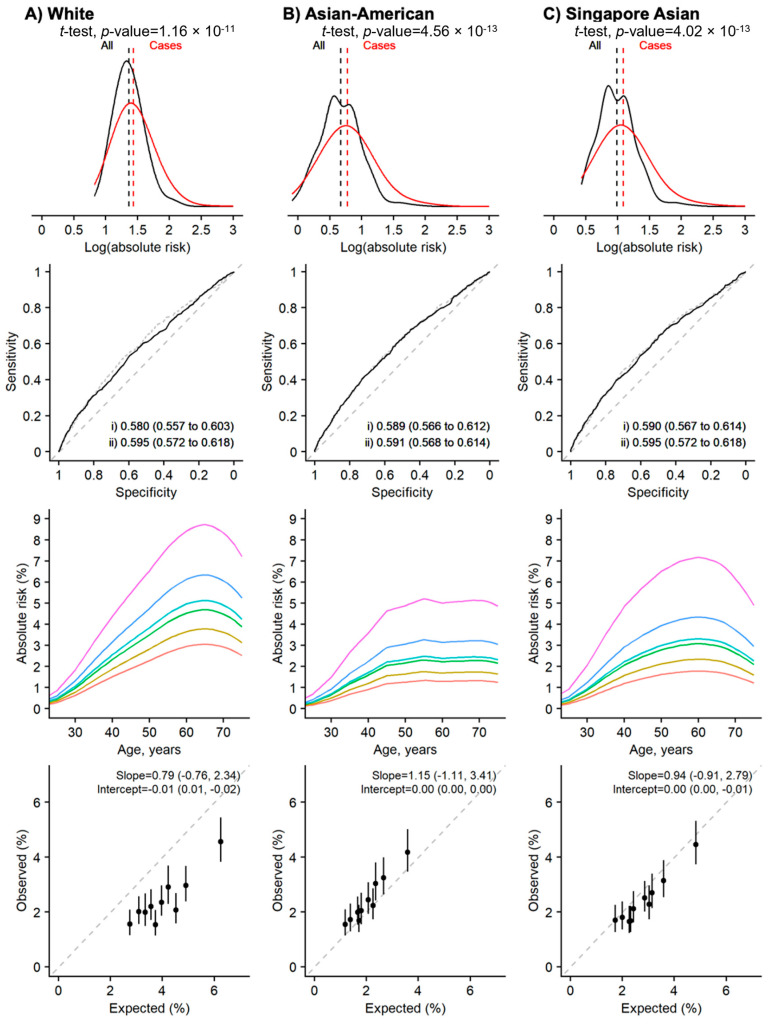
The Gail model (15-year absolute risk) performance assessment. Using the relative risk estimates from the White population [column **A**], and the closest Asian populations [columns **B** and **C**]. The BCRA package’s breast cancer incidence rates and mortality rates were replaced with the rates from Singapore’s population for the right-most panel [column **C**]. Rows (from top): (1) distribution, (2) discrimination, (3) predictive ability, and (4) calibration by deciles. Two-sided, two-sample *t*-tests with a type I error of 0.05 were used to examine whether there was a difference in the distribution of the log absolute risk between breast cancer cases and the entire population (row 1, the red and black lines are for cases and non-cases respectively). The Area Under the Receiver Operator Characteristic Curve (AUC) values: (i) unadjusted (solid line), and (ii) adjusted for age at recruitment (dashed line) (row 2). The colored lines (row 3, from bottom to top) in the plots for predictive ability denote 1-, 20-, 60-, 80-, 95-, and 99-percentile of the absolute risk. Calibration is calculated based on 15-year absolute risk by deciles (row 4).

**Figure 3 cancers-15-02559-f003:**
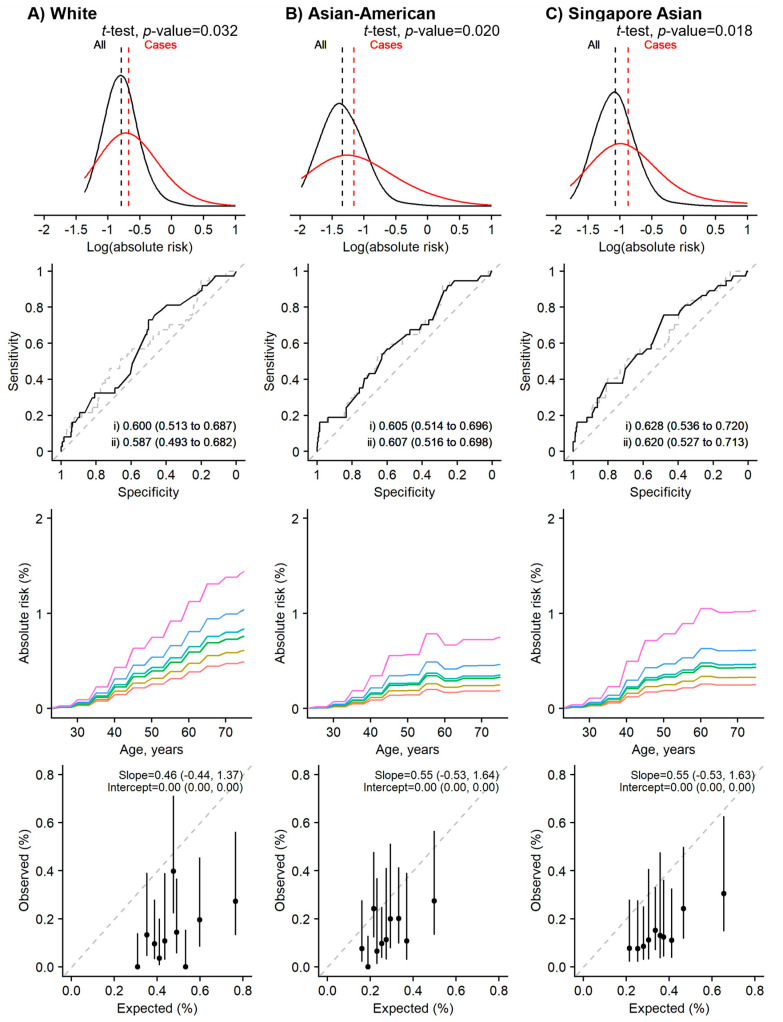
The Gail model (2-year absolute risk) performance assessment. Using the relative risk estimates from the White population [column **A**], and the closest Asian populations [columns **B** and **C**]. The BCRA package’s breast cancer incidence rates and mortality rates were replaced with the rates from Singapore’s population for the right-most panel [column **C**]. Rows (from top): (1) distribution, (2) discrimination, (3) predictive ability, and (4) calibration by deciles. Two-sided, two-sample *t*-tests with a type I error of 0.05 were used to examine whether there was a difference in the distribution of the log absolute risk between breast cancer cases and the entire population (row 1, the red and black lines are for cases and non-cases respectively). The Area Under the Receiver Operator Characteristic Curve (AUC) values: (i) unadjusted (solid line), and (ii) adjusted for age at recruitment (dashed line) (row 2). The colored lines (row 3, from bottom to top) in the plots for predictive ability denote 1-, 20-, 60-, 80-, 95-, and 99-percentile of the absolute risk. Calibration is calculated based on 2-year absolute risk by deciles in (row 4).

**Table 1 cancers-15-02559-t001:** Characteristics of the cohort, overall, and by breast cancer diagnosis before age 80 years. Absolute risk estimates for Whites and Asian-Americans are from the R package BRCA. The Chi-square test and Kruskal-Wallis test were used for categorical and continuous variables, respectively.

		Breast Cancer Occurrence before Age 80 Years		
	All (*n* = 26,380)	No (*n* = 25,380)	Yes (*n* = 1000)	*p*-Value
Age at screen, years (IQR)	57 (54–61)	57 (54–61)	56 (54–59)	4.16 × 10^−10^
Ethnicity (%)				
Chinese	22,208 (84)	21,351 (84)	857 (86)	1.51 × 10^−2^
Malay	1480 (6)	1445 (6)	35 (4)	
Indian	1317 (5)	1258 (5)	59 (6)	
Other	1375 (5)	1326 (5)	49 (5)	
Age at menarche, years (%)				
14+	16,934 (64)	16,351 (64)	583 (58)	1.05 × 10^−4^
12 to 13	8563 (32)	8193 (32)	370 (37)	
<12	872 (3)	825 (3)	47 (5)	
Missing	11 (0)	11 (0)	0 (0)	
Menopausal status, age in years (%)				
Pre-menopausal	2922 (11)	2781 (11)	141 (14)	7.61 × 10^−4^
Post-menopausal, <50	9571 (36)	9259 (36)	312 (31)	
Post-menopausal, 50 to 54	11,737 (44)	11,274 (44)	463 (46)	
Post-menopausal, 55+	2137 (8)	2053 (8)	84 (8)	
Missing	13 (0)	13 (0)	0 (0)	
Age at first life birth, years (%)				
<20	4429 (17)	4310 (17)	119 (12)	2.57 × 10^−12^
20 to 24	9778 (37)	9468 (37)	310 (31)	
25 to 29	7023 (27)	6731 (27)	292 (29)	
30+	3041 (12)	2880 (11)	161 (16)	
Nulliparous	1935 (7)	1831 (7)	104 (10)	
Missing	174 (1)	160 (1)	14 (1)	
Parity (%)				
Nulliparous	1935 (7)	1831 (7)	104 (10)	2.55 × 10^−7^
1 to 2	4200 (16)	4001 (16)	199 (20)	
3+	20,245 (77)	19,548 (77)	697 (70)	
Number of first degree relative with breast cancer (%)				
None	25,691 (97)	24,744 (97)	947 (95)	1.09 × 10^−9^
1	678 (3)	628 (2)	50 (5)	
2	11 (0)	8 (0)	3 (0)	
Recall status (%)				
Yes	1992 (8)	1851 (7)	141 (14)	2.19 × 10^−15^
None	24,388 (92)	23,529 (93)	859 (86)	
Biopsy ever (%)				
No	26,095 (99)	25,116 (99)	979 (98)	2.50 × 10^−3^
Yes	285 (1)	264 (1)	21 (2)	
Body mass index, kg/m^2^ (IQR)	24 (22–27)	24 (22–27)	25 (23–28)	3.32 × 10^−6^
Absolute risk, estimated from Whites (IQR)				
2-year	0.4 (0.4–0.5)	0.4 (0.4–0.5)	0.5 (0.4–0.5)	4.42 × 10^−5^
5-year	1.2 (1.0–1.4)	1.2 (1.0–1.4)	1.2 (1.0–1.4)	9.57 × 10^−5^
10-year	2.5 (2.1–2.9)	2.5 (2.1–2.9)	2.6 (2.2–3.1)	6.09 × 10^−7^
15-year	3.8 (3.3–4.5)	3.8 (3.3–4.5)	4.0 (3.5–4.7)	5.16 × 10^−10^
At age 80	5.9 (4.9–7.1)	5.9 (4.9–7.1)	6.6 (5.4–7.6)	2.87 × 10^−24^
Absolute risk, estimated from Asian-American (IQR)				
2-year	0.3 (0.2–0.3)	0.3 (0.2–0.3)	0.3 (0.2–0.3)	2.71 × 10^−15^
5-year	0.6 (0.5–0.8)	0.6 (0.5–0.8)	0.7 (0.6–0.9)	3.45 × 10^−17^
10-year	1.3 (1.1–1.6)	1.2 (1.1–1.6)	1.5 (1.2–1.7)	6.54 × 10^−18^
15-year	1.9 (1.7–2.4)	1.8 (1.7–2.3)	2.2 (1.7–2.5)	5.94 × 10^−18^
At age 80	2.9 (2.2–3.8)	2.9 (2.2–3.8)	3.3 (2.5–4.1)	3.73 × 10^−23^
Absolute risk, estimated from Singapore (IQR)				
2-year	0.3 (0.3–0.4)	0.3 (0.3–0.4)	0.4 (0.3–0.4)	3.90 × 10^−9^
5-year	0.9 (0.7–1.1)	0.9 (0.7–1.1)	0.9 (0.7–1.1)	2.64 × 10^−11^
10-year	1.7 (1.5–2.1)	1.7 (1.5–2.1)	2.0 (1.6–2.2)	8.92 × 10^−14^
15-year	2.5 (2.3–3.1)	2.5 (2.3–3.1)	3.0 (2.3–3.3)	4.86 × 10^−16^
At age 80	4.0 (3.1–5.0)	3.9 (3.1–5.0)	4.5 (3.4–5.4)	7.26 × 10^−25^

**Table 2 cancers-15-02559-t002:** Association between the Gail model absolute risk and the age at diagnosis in women who developed breast cancer within the expected years (i.e., using the 2-year absolute risk to predict the age of diagnosis of those who developed breast cancer within 2 years of screening), using linear models. RR refers to the relative risk estimates from the BCRA package (“White” or “Asian-American” population). * Adjusted for age at screening and ethnicity. SE: standard error.

		Unadjusted			Adjusted *		
Parameters (RR/Incidence and Mortality Rates)	X-Year Absolute Risk	Beta	SE	*p*-Value	Beta	SE	*p*-Value
White/White	2	−1.26	3.47	0.720	0.05	0.56	0.929
	5	1.36	0.73	0.065	−0.10	0.26	0.694
	10	0.52	0.22	0.019	−0.03	0.14	0.813
	15	0.06	0.16	0.686	−0.15	0.12	0.197
Asian-American/Asian-American	2	−7.82	3.05	0.015	−0.09	0.62	0.887
	5	−2.59	0.77	0.001	−0.30	0.29	0.295
	10	−1.36	0.33	<0.001	−0.34	0.21	0.106
	15	−0.88	0.22	<0.001	−0.20	0.17	0.235
Asian-American/Singapore	2	−2.48	1.64	0.139	0.15	0.27	0.592
	5	−0.89	0.51	0.082	−0.23	0.18	0.197
	10	−0.47	0.2	0.022	−0.21	0.12	0.097
	15	−0.43	0.14	0.003	−0.17	0.11	0.109

**Table 3 cancers-15-02559-t003:** Absolute risk of developing breast cancer for an average 50-year-old woman in Singapore. Higher-than-average risk was defined as the absolute risk above the 60th percentile of the study population (i.e., above-average risk threshold). RR refers to the relative risk estimates associated with breast cancer risk factors from the BCRA package (“White” or “Asian-American” population). * (Highest − Lowest)/Highest × 100.

X-Year Absolute Risk	Parameters (RR/Incidence and Mortality Rates)	Absolute Risk (%)	Percentage Difference between the Highest and Lowest Threshold for X-Year Absolute Risk *
2	White/White	0.39	37
	Asian-American/Asian-American	0.25	
	Asian-American/Singapore	0.33	
5	White/White	0.98	37
	Asian-American/Asian-American	0.62	
	Asian-American/Singapore	0.82	
10	White/White	2.14	32
	Asian-American/Asian-American	1.46	
	Asian-American/Singapore	1.74	
15	White/White	3.50	38
	Asian-American/Asian-American	2.16	
	Asian-American/Singapore	2.79	

## Data Availability

Third-party data access of the dataset (Y21-S0020-1) used in this study are available from the National Registry of Diseases Office (NRDO) upon request (https://nrdo.gov.sg/data-request, accessed on 10 May 2022). For access to confidential information, interested researchers must submit requests for data to Dr. Fuh Yong Wong (wong.fuh.yong@singhealth.com.sg) and Dr. Li Jingmei (lijm1@gis.a-star.edu.sg).

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
