# Peer review of "Will Absolute Risk Estimation for Time to Next Screen Work for an Asian Mammography Screening Population?"

_cancers, 2023, doi:10.3390/cancers15092559_

Round 1

Reviewer 1 Report

Authors examined the applicability of the Gail model, a well-known breast cancer prediction tool including several risk factors, in an Asian population. The results show that the tool performs better over a longer prediction period, but it is not yet possible to identify women with increased risk in the short term, so the importance of routine screening should be emphasized.This article is very innovative. I don't have any more comments.

Author Response

Authors examined the applicability of the Gail model, a well-known breast cancer prediction tool including several risk factors, in an Asian population. The results show that the tool performs better over a longer prediction period, but it is not yet possible to identify women with increased risk in the short term, so the importance of routine screening should be emphasized. This article is very innovative. I don't have any more comments.

Our response: Thank you.

Reviewer 2 Report

Dear Authors, 

first of all, congratulations for your interesting work. I hope that my hints will help you in the next steps of improvement and the final manuscript will be really valuable for the readers. Very good discussion section, well done. 

General comments:

 - please double-check the sources, in several sentences the sources are missing.

- capital letters and italics are sometimes missed/overused, as well as several typos and punctation mistakes. Please correct.

Specific comments:

Line 47 --> I believe mammography instead of mammogram?

Lines 49-52 --> There are some new papers showing cost-effectiveness of regular mammography, highlighting also the importance of personalisation (risk factors should be taken into account). It might be interesting to mention also this approach in your introduction - since financial angle is extremely important in public health systems.

Line 75 --> and for which population/s it works best?

Line 100 --> this is a serious drawback of the paper; it's been over 30 years ago and many factors have changed since that moment, including the quality of the mammography devices. I would hesitate it this study is still relevant. Or have you followed the patients from that moment until today? If yes, then this should be clearly indicated in the paper. 

- Some minor English language improvements should be applied.

 - please double-check the sources, in several sentences the sources are missing.

- capital letters and italics are sometimes missed/overused, as well as several typos and punctation mistakes. Please correct.

Author Response

First of all, congratulations for your interesting work. I hope that my hints will help you in the next steps
of improvement and the final manuscript will be really valuable for the readers. Very good discussion section, well done.

Our response: Thank you.

General comments:

 - please double-check the sources, in several sentences the sources are missing.

Our response: We have added references where they are missing to the best of our ability.

- capital letters and italics are sometimes missed/overused, as well as several typos and punctation mistakes. Please correct.

Our response: We have reviewed spelling, grammar, punctuation, clarity, engagement, and delivery mistakes using a digital proof-reader and made the suggested edits.

Specific comments:

Line 47 --> I believe mammography instead of mammogram?

Our response: Edited as per suggested to “An effective strategy for mortality reduction is to routinely screen asymptomatic women using mammography from the age of 50 years.”

Lines 49-52 --> There are some new papers showing cost-effectiveness of regular mammography, highlighting also the importance of personalisation (risk factors should be taken into account). It might be interesting to mention also this approach in your introduction - since financial angle is extremely important in public health systems.

Our response: “In terms of cost-effectiveness, age-targeted regular mammography screening for breast cancer is reported to be cost-effective, even when considering false positives and overdiagnosis (10.7326/0003-4819-148-1-200801010-00002). However, considering a woman's individual breast cancer risk factors can potentially improve the cost-effectiveness of mammography screening (10.1371/journal.pone.0226352). Breast cancer risk prediction models thus present an opportunity in nationwide risk-based breast cancer screening programs by identifying individuals at high risk who may benefit more from interventions [7-9].”

Line 75 --> and for which population/s it works best?

Our response: We have added in a sentence to explain that the Gail model works well in populations of European ancestry: “The model has been reported to perform well in populations of European ancestry (10.1186/s13058-018-0947-5). However, it is not well-calibrated for Asian populations [22, 23].”

Line 100 --> this is a serious drawback of the paper; it's been over 30 years ago and many factors have changed since that moment, including the quality of the mammography devices. I would hesitate it this study is still relevant. Or have you followed the patients from that moment until today? If yes, then this should be clearly indicated in the paper.

Our response: We have added a statement to explain that passive follow-up is on-going:

“As part of this prospective cohort, 28,234 were recruited between October 1994 and February 1997. Passive follow-up on the cohort was performed using national registries.

In section 2.3, it was explained that “The first diagnosis of invasive breast cancer, occurring at least 6 months after recruitment and before age 80 years, was identified via linkage with the Singapore Cancer Registry with the latest date of occurrence set at 31 December 2019.”

- Some minor English language improvements should be applied.

Our response: We have reviewed spelling, grammar, punctuation, clarity, engagement, and delivery mistakes using a digital proof-reader.

 - please double-check the sources, in several sentences the sources are missing.

Our response: We have added references where they are missing to the best of our ability.

- capital letters and italics are sometimes missed/overused, as well as several typos and punctuation mistakes. Please correct.

Our response: We have reviewed spelling, grammar, punctuation, clarity, engagement, and delivery mistakes using a digital proof-reader and made the suggested edits.